# Aerosol-generating behaviours in speech pathology clinical practice: A systematic literature review

**Antonia Margarita Chacon**[1,2]*, **Duy Duong Nguyen**[1,2], **Patricia McCabe**[1], **Catherine Madill**[1,2]

1 Discipline of Speech Pathology, Faculty of Medicine and Health, Sydney School of Health Sciences, The University of Sydney, Sydney, Australia, 2 Doctor Liang Voice Program, Faculty of Medicine and Health, Sydney School of Health Sciences, The University of Sydney, Sydney, Australia

* antonia.chacon@sydney.edu.au

## Abstract

### Objective

To evaluate the evidence of aerosol generation across tasks involved in voice and speech assessment and intervention, to inform better management and to reduce transmission risk of such diseases as COVID-19 in healthcare settings and the wider community.

### Design

Systematic literature review.

### Data sources and eligibility

Medline, Embase, Scopus, Web of Science, CINAHL, PubMed Central and grey literature through ProQuest, The Centre for Evidence-Based Medicine, COVID-Evidence and speech pathology national bodies were searched up until August 13th, 2020 for articles examining the aerosol-generating activities in clinical voice and speech assessment and intervention within speech pathology.

### Results

Of the 8288 results found, 39 studies were included for data extraction and analysis. Included articles were classified into one of three categories: research studies, review articles or clinical guidelines. Data extraction followed appropriate protocols depending on the classification of each article (e.g. PRISMA for review articles). Articles were assessed for risk of bias and certainty of evidence using the GRADE system. Six behaviours were identified as aerosol generating. These were classified into three categories: vegetative acts (coughing, breathing), verbal communication activities of daily living (speaking, loud voicing), and performance-based tasks (singing, sustained phonation). Certainty of evidence ranged from very low to moderate with variation in research design and variables.

**Competing interests:** The authors have declared that no competing interests exist.

## Conclusions

This body of literature helped to both identify and categorise the aerosol-generating behaviours involved in speech pathology clinical practice and confirm the low level of evidence throughout the speech pathology literature pertaining to aerosol generation. As many aerosol-generating behaviours are common human behaviours, these findings can be applied across healthcare and community settings.

## Systematic review registration

Registration number CRD42020186902 with PROSPERO International Prospective Register for Systematic Reviews.

## Introduction

Aerosol generation (or aerosolisation) is the suspension in air of liquid or solid particles [1]. Aerosols form part of a continuum of particle sizes generated through a range of respiratory activities in humans, which can carry infective viral material and facilitate respiratory disease transmission [2]. Aerosol-generating procedures (AGPs) have been defined in the literature as medical procedures which lead to the generation of aerosols of sufficient size to enable viral transmission [3]. Despite this restricted consideration of AGPs as occurring in medicine, there is now an imperative to investigate what is known about other human aerosolising activities or aerosol-generating behaviours (AGBs) that occur in other healthcare settings and by extension, in everyday life.

Severe acute respiratory syndrome coronavirus 2 (SARS-CoV-2) or COVID-19 is a highly contagious virus [4] with a high mortality rate [5] and emerging morbidity associated with long COVID [6], which has resulted in an enormous burden upon the healthcare system globally [7, 8]. Healthcare workers (HCWs) face an especially high risk of contracting COVID-19 in their patient interactions, owing to the ease of virus transmission through airborne aerosols, droplets and fomite contact [8, 9] and preliminary evidence that the highest viral load of coronavirus in the body is localised in the oro- and nasopharynx, with sputum and nasal/throat swabs containing significantly higher traces of the virus than samples taken from other bodily fluids, such as blood and urine [10, 11].

Speech language pathology (SLP) is an area of health care dedicated to the assessment and treatment of communication and swallowing disorders. Speech language pathologists assess and treat both children and adults with voice, resonance, speech, language, fluency and swallowing disorders (dysphagia) across a wide range of clinical and professional populations. They use a variety of laryngeal, voice and speech tasks in the assessment and treatment of voice, speech, and swallowing dysfunction. These include testing vegetative reflexes/functions (e.g. coughing and breathing), verbal activities of daily living (e.g. conversational speech, loud voicing, standard phrases [12], reading passages [13]) and performance tasks (sustained vowel phonation, singing). A common feature of assessment and treatment tasks is the use of respiratory airflow in combination with a range of laryngeal manoeuvres and/or vocal tract/articulatory movements, which can be sources for aerosol generation [14]. Many of these tasks also occur commonly in healthcare settings and daily face-to-face societal interactions. The Royal College of Speech and Language Therapists (RCSLT) [15] and an additional speech pathology research group [16] have respectively produced reviews of the literature regarding COVID-19

and dysphagia; however to date, no research has been conducted to explore the aerosol transmission risk associated with COVID-19 in other areas of SLP clinical practice. As COVID-19 has been found to spread via aerosols generated during such respiratory activities as breathing, speaking, and coughing, SLPs appear to be at a particularly high risk of disease contraction when working with COVID-positive patients. It is therefore critical to understand the aerosol-generating potential of these and other phonatory and speech tasks, as they may impact all people involved in person-to-person contact where verbal communication takes place.

This review seeks to determine what evidence exists of AGPs in an office-based SLP setting, which in turn, may help inform the processes and procedures that may be implemented for clinicians to safeguard their patients and themselves in the assessment and management of voice, resonance and motor speech disorders. The findings of this review will have implications for HCWs more broadly, especially in person-to-person communication-focused contexts such as medical examinations, counselling, rehabilitation and aged care. As the activities investigated are also activities of daily living, the findings are highly relevant to the community at large.

## Methods

### Protocol and registration

The systematic review was conducted according to the Preferred Reporting Items for Systematic Reviews and Meta-analyses (PRISMA) [17] and Synthesis Without Meta-analysis (SWiM) in Systematic Reviews reporting guidelines [18]. The SWiM guideline was used as a means of facilitating synthesis and promoting clear reporting of the findings. Owing to the diversity of study characteristics and inconsistency in reporting effect estimates across the extracted articles, this tool was deemed to be an appropriate checklist. The protocol was registered through the PROSPERO International Prospective Register for Systematic Reviews (registration number CRD42020186902).

### Information sources

Databases searched were Medline (OVID interface), Embase (OVID interface), Scopus, Web of Science, CINAHL and PubMed Central. Grey literature was also searched through ProQuest to capture unpublished dissertations, The Centre for Evidence-Based Medicine (CEBM), COVID-Evidence and speech pathology peak national bodies inclusive of Speech Pathology Australia (SPA), The Royal College of Speech and Language Therapists (RCSLT) and The American Speech Language Hearing Association (ASHA).

### Search strategy

The initial search was conducted by the first author (AC) on 22 May 2020 and limited to articles published after January 1940; the earliest date found for a relevant article when conducting a pre-study scoping review. The first author conducted a final search to include new articles published to 13 August 2020. The search strategy was initially determined through discussions between three authors (AC, CM and PM) and in accordance with the findings of the initial scoping review. It was then further developed in collaboration with a senior research librarian specialised in the area of health sciences, with expertise in systematic review searching.

The search string consisted of terms that were grouped according to concepts being relevant to speech pathology, COVID-19, AGPs and the areas of voice, resonance and speech. AGBs was not used as this term had not yet been published in the literature at the time of this study. Within the selected 'concept areas' we developed a list of synonyms and/or specific terms relevant to our search scope; being coronavirus, speech pathology, speech pathology

clinical tasks, aerosols and transmission risk. The terms associated with each concept area were systematically searched against other concept word lists to ensure literature saturation of all relevant articles. An exemplar search strategy applied to the Medline, Embase, Scopus, Web of Science, CINAHL and PubMed Central databases can be found in the S2 File.

## Inclusion criteria

The study search plan initially focused upon articles examining the AGPs involved in speech pathology voice, resonance and motor speech clinical tasks, as well as recommendations on how to reduce COVID-19 transmission risk for these procedures. Owing to the complexity of the results recovered from conducting a search across these two areas (i.e. AGP classification and recommendations), the search strategy was refined to only focus on AGPs in speech pathology clinical practice.

Studies and unpublished works were included if they focused on human subjects over the age of 12 years, involved an outpatient or clinic office setting, were relevant to the conduction of speech pathology voice, resonance and motor speech clinical tasks, and discussed AGPs in English. Articles were excluded if the focus population was under 12 years of age, did not explore the behaviours of breathing, coughing, speaking, singing, loud voicing or sustained phonation, were set exclusively in an inpatient hospital setting, explored ENT-specific or invasive procedures only, were based on animal studies, and/or were not written in the English language. Additional exclusion criteria applied to published works included publications which were not peer-reviewed and/or did not cite any references.

## Study records

The initial database searches retrieved 7,724 records. Searches of the abovementioned grey literature sources and unpublished studies (including practice guidelines, unpublished dissertations, government and organisational reports) were also completed at this stage, identifying 564 additional articles that appeared to meet the inclusion criteria. We therefore collected a total of 8,288 records. These records were uploaded to the Covidence platform (www. covidence.org) to manage data, facilitate collaboration and document the review process over the course of the study.

Covidence identified 2,195 duplicates which were then removed for a total of 6,093 records. Titles and abstracts were screened against the inclusion criteria by two independent reviewers (any combination of AC, CM, PM and DN). Any disagreements that arose between the reviewers at each stage of the selection process were resolved through the involvement of a third reviewer. Five thousand, nine hundred and thirty-three records were excluded based on titles and abstracts. Full texts of the remaining 160 records were assessed in detail against the inclusion criteria by two independent reviewers (AC, CM). Articles that did not meet the study criteria were removed, with reasons for exclusion being recorded. One hundred and twenty-one studies were excluded from this process (see PRISMA flow diagram). A further hand search of the remaining articles' reference lists, and any articles published to 13 August 2020 was conducted (AC). Following a further process of title/abstract screening (AC, CM), full text review and exclusion of inappropriate studies (CM, DN), an additional 21 studies were included. The final systematic review included 39 studies.

## Data extraction and data items

Data was extracted from included papers by two independent reviewers (AC, DN). Three data extraction tables were used according to the type of paper from which data was being extracted; being research studies, review articles and clinical guidelines. The STROBE checklist

[19], was used to extract data from research studies, the PRISMA guidelines [17] were used to extract data from review articles and the RIGHT checklist [20] was used to extract data from clinical guideline documents. A simplified version of the data extraction tables is presented in the S3 File. Quantitative synthesis and meta-analyses were not completed owing to the heterogeneity of data and designs across studies.

### Identification and characterisation of AGPs

All eligible research studies, review articles and clinical guidelines that measured aerosol characteristics related to risk of infectious disease transmission in speech pathology voice, resonance and motor speech assessment and treatment tasks were included. Data extracted included variables used to characterise identified AGPs (including size and distribution of aerosol particles and dynamic characteristics of aerosols) and specific voice and speech tasks associated with aerosol generation.

### Evidence for risk of infectious disease transmission associated with AGPs

Data pertinent to the risk and mechanism of infectious disease transmission of identified AGPs were also extracted. This included information pertaining to the presence of viable pathogens in aerosols and spreading capability of aerosols across the identified AGPs.

### Evaluation of certainty of evidence and risk of bias

The original study protocol specified that the Cochrane Risk of Bias Tool would be used to assess risk of bias in individual studies. However, owing to the diversity of article types collated throughout the review process, it was decided that a more holistic tool examining the overall quality of each article, including risk of bias, would provide a more consistent and rigorous evaluation across all 39 studies.

The certainty of the included evidence was assessed through the Grading of Recommendations Assessment, Development and Evaluation (GRADE) working group methodology [21]. This involved examining the quality of evidence through the domains of risk of bias, consistency, precision, directness and publication bias. Following this evaluation, the first author (AC) determined whether the quality of the research could be deemed as high (i.e. very unlikely that further research will change our confidence in the estimate of effect), moderate (i.e. likely that further research will have an impact on our confidence in the estimate of effect and may change the estimate), low (i.e. very likely that further research will have an important impact on our confidence in the estimate of effect and is likely to change the estimate), or very low (i.e. very uncertain about the estimate of effect). The GRADEpro app was used to facilitate this process and ensure that the abovementioned terms were informed by a consistent, systematic process [22–24].

## Results

### Identified studies

The Fig 1 PRISMA diagram outlines the process undertaken to collect and review published and unpublished records. Thirty-nine records were identified as meeting the inclusion criteria for the review. Twenty-four records were classified as original research studies, 12 were classified as review articles and three were classified as clinical guidelines.

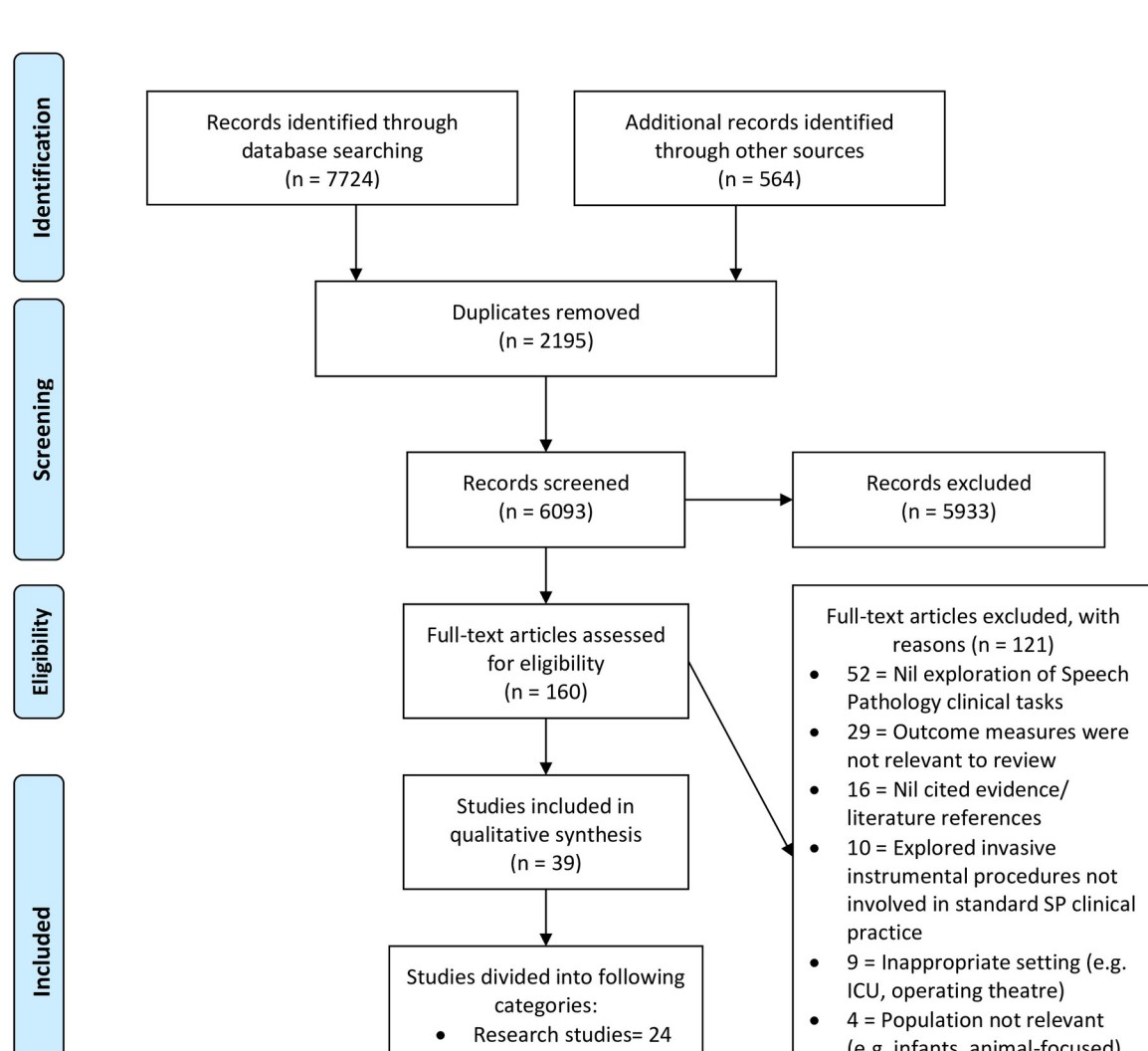

**Fig 1. PRISMA flowchart of study inclusions and exclusions.** *From*: Moher D, Liberati A, Tetzlaff J, Altman DG, The PRISMA Group (2009). *Preferred Reporting Items for Systematic Reviews and Meta-Analyses: The PRISMA Statement. PLoS Med 6(7): e1000097. doi: 10. 1371/journal.pmed1000097* **For more information, visit** www.prisma-statement.org.

## Study characteristics

**Research studies.** Two research studies were published before the year of 2000; 9 were published from 2006–2010, and 13 were published between 2011–2020. Twenty studies used an experimental design using human participants and one study used available human data to evaluate the risk of transmission [25]. Three studies were based on modelling or mathematical designs. There were no randomised clinical trial (RCT) nor prospective cohort studies. For

studies using human subjects, sample size ranged between 1 and 61. Two studies did not report sample size.

**Review articles.** Of the review papers, two were published in or before 2010, two were published between 2011–2019, and eight were published in 2020. There were seven reviews, two rapid reviews, one scoping review, one narrative review and one fact sheet. There were no systematic reviews.

**Clinical guidelines.** All three clinical guidelines were published in 2020.

## Identification and characterisation of AGPs

Table 1 presents an overview of each record included in the review, providing a summary of settings, AGPs and specific virus types explored. The data extracted from each record is summarised in tabular format as a S3 File. Table 2 presents all identified AGPs with the highest GRADE level of evidence across the studies within each AGP category. Coughing was examined in the largest number of studies (31 records), followed by speaking (21 records), breathing (16 records), singing (4 records), sustained phonation (3 records), and loud voicing (3 records). The levels of evidence for each AGP varied greatly across studies.

Table 3 describes the diversity in research focus explored across the collated studies and AGP areas. While certain papers explored specific characteristics of aerosol particles (e.g. diameter), others placed a greater focus upon the dynamics of aerosol movement (e.g. velocity, air dispersion).

**Coughing.** Fourteen research studies were conducted on aerosol generation when coughing. The four studies of moderate evidence explored the characteristics of air dispersion distance, particle size distribution and particle volume and concentration. The studies of moderate evidence revealed that 82% of the droplet nuclei produced when coughing are within a sufficiently small range (0.74–2.12 micrometres, μm) to contribute to airborne disease transmission [33]. Morawska et al also found that a large proportion of coughed droplet nuclei fall below the 0.8 μm diameter range [32]. Two studies revealed that a greater number and volume of particles are expelled per cough in individuals with influenza as compared to those without [31], and that the mean surface area of these particles is greater when participants are infected with influenza compared to when they are well [30]. The three studies of lower evidence explored the above characteristics, in addition to airflow dynamics, airflow imaging and duration of air carriage. Overall, coughing can generate aerosols of a sufficiently small size (0.74–2.12 μm) to enable disease transmission, and increased aerosol generation occurs when a participant is infected with influenza, as opposed to when healthy.

**Speaking.** Nine research studies explored speaking as an AGP. Specific speech tasks associated with aerosol generation were explored in several of the included research studies. This involved vowel production in two studies [28, 41], voiced consonants in two studies [28, 41], voiceless consonants in two studies [28, 41], voiced plosive consonants in one study [28], and voiceless fricatives in one study [28].

The three studies of moderate evidence explored particle emission rate, concentration and size. These studies revealed a relationship between particle release and vocalisation and identified that there is substantial between-subject variability in particle emission rate [27]. Environmental factors of temperature and humidity were reported to have no impact on emission rate nor particle size, and a higher emission rate was found to exist for speech than breathing [27]. A further study of moderate evidence found that particular sounds (e.g. /i/) produce more particles than others (e.g. /a/), the volume of particles produced was greater in words with voiced plosive consonants than voiceless fricatives and that the rate of particle emission positively

**Table 1. Summary of characteristics of included studies.**

| Record category | Study name | Study design/ Review Type | Target setting/s explored | AGPs investigated related to SP practice | Virus examined | Certainty of Evidence (GRADE rating) |
|---|---|---|---|---|---|---|
| Research studies | Adhikari et al., 2019 [26] | Case study | Hospital | Coughing | MERS | Moderate |
| | Asadi et al., 2019 [27] | Cross-sectional | Not specified (lab-based) | Breathing, Speaking | Nil specific | Moderate |
| | Asadi et al., 2020 [28] | Cross-sectional | Not specified (lab-based) | Speaking | Nil specific | Moderate |
| | Holmgren et al., 2010 [29] | Cross-sectional | Not specified (lab-based) | Breathing (tidal and airway closure manoeuvre) | Nil specific | Moderate |
| | Lee et al., 2018 [30] | Cohort | Not specified (lab-based) | Coughing | Influenza | Moderate |
| | Lindsley et al., 2012 [31] | Cohort | Not specified (lab-based) | Coughing | Influenza | Moderate |
| | Morawska et al. 2009 [32] | Cross-sectional | Not specified (lab-based) | Breathing, Coughing, Sustained vowel phonation, Speaking, Whispering, | Nil specific | Moderate |
| | Yang et al., 2007 [33] | Cross-sectional | Not specified (lab-based) | Coughing | Nil specific | Moderate |
| | Johnson & Morawska, 2009 [34] | Cross-sectional | Not specified (lab-based) | Breathing | Nil specific | Low |
| | Johnson et al., 2011 [35] | Cross-sectional | Not specified (lab-based) | Breathing, Coughing, Sustained vowel phonation, Speaking | Nil specific | Low |
| | Lindsley et al., 2016 [36] | Cross-sectional | Not specified (lab-based) | Breathing (exhalation), Coughing | Influenza | Low |
| | Stelzer-Braid et al., 2009 [37] | Cross-sectional | Not specified (lab-based) | Breathing, Coughing, Speaking | Rhinovirus, parainfluenza, influenza, human metapneumovirus | Low |
| | You et al., 2013 [38] | Survey and cross-sectional | High school | Coughing, Speaking | Nil specific | Low |
| | Zayas et al., 2012 [39] | Cross-sectional open bench | Not specified (lab-based) | Coughing | Nil specific | Low |
| | Duguid, 1946 [40] | Cross-sectional | Not specified (lab-based) | Coughing, Speaking | Nil specific | Very low |
| | Georgiou & Kilani, 2020 [25] | Cross-sectional | Not specified (lab-based) | Speaking | COVID-19 | Very low |
| | Giovanni et al., 2020 [41] | Cross-sectional | Not specified (lab-based) | Breathing, Sustained vowel phonation, Speaking (voiced and voiceless fricative consonants; reading), Semi-occluded vocal tract (SOVT) exercises | COVID-19 | Very low |
| | Hui et al., 2012 [42] | Cross-sectional | Hospital | Coughing | Nil specific | Very low |
| | Nicas & Jones, 2009 [43] | Cross-sectional | Hospital/ Long-term care setting, Residential bedroom | Coughing | Influenza | Very low |
| | Papineni & Rosenthal, 1997 [44] | Cross-sectional | Not specified (lab-based) | Coughing, Nose breathing, Mouth breathing, Speaking, | Nil specific | Very low |
| | Tang et al., 2009 [45] | Observational | Not specified (lab-based) | Coughing | Nil specific | Very low |
| | Tang et al., 2012 [46] | Cross-sectional | Not specified (lab-based) | Coughing | Nil specific | Very low |
| | Xie et al, 2009 [47] | Cross-sectional | Not specified (lab-based) | Coughing, Speaking | Nil specific | Very low |
| | Zhu et al, 2006 [48] | Cross-sectional | Office and bedroom spaces simulated | Coughing | Influenza | Very low |

*(Continued)*

**Table 1.** (Continued)

| Record category | Study name | Study design/ Review Type | Target setting/s explored | AGPs investigated related to SP practice | Virus examined | Certainty of Evidence (GRADE rating) |
|---|---|---|---|---|---|---|
| Review articles | Gralton et al., 2011 [49] | Not specified beyond 'review article' | Not specified | Breathing, Coughing, Speaking | Not specified | Low |
| | Quereshi et al, 2020 [50] | Rapid review | Hospital and community-based settings | Not specific to SP AGPs, however mention of breathing and coughing | COVID-19 | Low |
| | Tang et al., 2006 [51] | Not specified beyond 'review article' | Hospitals, clinics | Coughing, Speaking | SARS (2003 outbreak) | Low |
| | Wilson et al., 2020 [52] | Narrative review | Not specified (however clinic and hospital settings named) | Breathing (dyspnoeic), Coughing | COVID-19 | Low |
| | Zemouri et al., 2017 [53] | Systematic scoping review | Hospitals, clinics | Breathing, Coughing, Speaking | Not specified | Low |
| | Bolton et al., 2020 [15] | Not specified; identified as 'review and report on evidence' | Hospital and clinic | Coughing, Speaking | COVID-19 | Very low |
| | Carlson et al., 2010 [1] | Not specified; described as 'overview' | Hospital, university, holiday tour (inside and outside bus) | Speaking | H1N1 influenza A (pH1N1) | Very low |
| | Mick & Murphy, 2020 [54] | Literature review | Hospital | Breathing (normal; pursed lip), Coughing, Speaking | COVID-19 | Very low |
| | Naunheim et al., 2020 [55] | Not specified beyond 'review article' | Rehearsal and performance spaces | Singing | COVID-19 | Very low |
| | Pasnick et al., 2020 [3] | Fact sheet | Not specified (however hospital setting suggested) | Coughing, Singing, Speaking | COVID-19 | Very low |
| | Viswanath et al., 2020 [56] | Rapid review | Hospital, clinic, laboratory | Coughing, Speaking | COVID-19 | Very low |
| | Xu et al., 2020 [57] | Not specified beyond 'review article' | Hospitals, clinics | Breathing, Coughing, Speaking in close contact | COVID-19 | Very low |
| Clinical guidelines | Mattei et al., 2020 [58] | Not applicable | Not specified, however applicable to hospital and clinic office environments | Coughing | COVID-19 | Very low |
| | RCSLT, 2020 [59] | Not applicable | SLP clinical work settings i.e. clinics, hospitals, etc. | Breathing; Coughing; Loud voicing; Singing | COVID-19 | Very low |
| | SPA, 2020 [60] | Not applicable | SLP clinical work settings i.e. clinics, hospitals, etc. | Coughing, Singing, Speaking, Voice Ax tasks | COVID-19 | Very low |

correlated with the vowel content of a phrase [28]. Finally, it was found that, compared with the activities of breathing, coughing, sustained phonation and whispering, the particles generated by speaking tend to involve a broader size distribution, including some particles that are larger in size than produced by other aerosol-generating behaviours i.e. in the 3.5 and 5 μm range [32]. The six research studies of lower evidence, in addition to these areas, also explored airflow dynamics and duration of air carriage.

Owing to the dynamic nature of speech, and the frequency with which humans engage in verbal communication, it appears that speaking emits more particles than non-speech breathing, however the size of particles produced when speaking tend toward a predominance of larger particles when compared to other AGPs.

**Breathing.** Seven research studies explored breathing as an AGP. Three studies on breathing were of moderate level evidence and investigated aerosol-generating characteristics of particle concentration and size. Nose breathing, mouth breathing, deep-fast breathing and fast-deep breathing produced a lower particle emission rate as opposed to speech, however

**Table 2. Types of AGPs identified.**

| AGP | Number of articles exploring AGP | Specific papers exploring AGP | Highest level of evidence (GRADE) |
|---|---|---|---|
| Coughing | 31 | [3, 15, 26, 30–33, 35–40, 42–54, 56–60] | Moderate |
| Speaking | 21 | [1, 3, 15, 25, 27, 28, 32, 35, 37, 38, 40, 41, 44, 47, 49, 51, 53, 54, 56, 57, 60] | Moderate |
| Breathing | 16 | [27, 29, 32, 34–37, 41, 44, 49, 50, 52–54, 57, 59] | Moderate |
| Singing | 4 | [3, 55, 59, 60] | Very low |
| Sustained phonation | 3 | [32, 35, 41] | Moderate |
| Loud voicing | 3 | [27, 59, 60] | Moderate |

individual variance was noted across participants [27]. There was no significant impact of neither temperature nor humidity on the emission rate nor size of emitted particles during breathing. A large proportion of particles produced during breathing were of diameters $< 0.8\mu m$, and the average particle concentrations produced during exhalation were $0.1/cm^3$ [32]. Investigation of aerosol generation in different manners of breathing found that the airway closure manoeuvre (i.e. exhaling slowly until the participant reaches residual volume) produced a significantly higher concentration of particles than tidal (i.e. 'normal') breathing [29]. Breathing with airway closure elicited the same particle mode size as was found in tidal breathing, however an additional stronger and broader maximum was found between

**Table 3. Characteristics of identified AGPs based on evidence from the included studies.**

| AGPs with Aerosol Properties explored | Number of articles exploring AGP aerosol properties | Area of focus | Specific papers exploring AGP |
|---|---|---|---|
| Coughing | 18 | Air dispersion distance | [30, 42] |
| | | Airflow dynamics | [38, 46, 48] |
| | | Airflow imaging | [45] |
| | | Classification of procedure as aerosol-generating | [15, 53, 54] |
| | | Duration of air carriage | [40] |
| | | Microbial load and composition of aerosols | [53] |
| | | Particle concentration/ number/ volume | [30, 31, 39, 44, 47] |
| | | Particle/ Droplet and droplet nuclei size and/or size distribution | [32, 33, 35, 39, 40, 44, 47, 49] |
| Speaking | 13 | Airflow dynamics | [38] |
| | | Classification of procedure as aerosol-generating | [15, 53, 54] |
| | | Duration of air carriage | [40] |
| | | Microbial load and composition of aerosols | [53] |
| | | Particle emission rate | [28] |
| | | Particle concentration/ number | [25, 27, 44, 47] |
| | | Particle size and/or distribution | [27, 32, 35, 40, 44, 47, 49] |
| Breathing | 10 | Classification of procedure as aerosol-generating | [53, 54] |
| | | Microbial load and composition of aerosols | [53] |
| | | Particle or aerosol concentration/ number | [27, 34, 44] |
| | | Particle/ aerosol size and /or distribution | [27, 29, 32, 34, 35, 44, 49] |
| | | Velocity of exhaled air | [41] |
| Sustained phonation | 3 | Droplet/ particle size distribution | [32, 35] |
| | | Velocity of exhaled air | [41] |

0.2–0.5μm. The further lower-evidence studies (2 low, 2 very low certainty of evidence) examining breathing as an AGP explored the same aerosol characteristics, in addition to the velocity of exhaled air. Overall, it was found that breathing emits small aerosol particle sizes (<0.8μm), of which size and concentration tend to be unaffected by such environmental factors as temperature and humidity. Particle emission rates for this AGP are lower than that for speech. Certain modalities of breathing may generate higher concentrations of aerosolised particles than other modes and as with other AGPs, substantial variation between subjects regarding particle size and concentration was reported.

**Sustained phonation.** Sustained phonation was explored in three research studies. The single study of moderate evidence identified the particle size and concentrations produced by this AGP, reporting that sustained phonation produced average concentrations of 0.04 and 0.16cm$^3$, and particles between 3.5 and 5 μm in size became more prominent in sustained phonation compared to speech and other explored AGPs [32]. The study of very low evidence additionally explored the characteristics of exhaled air velocity. The findings suggest that sustained phonation yields higher concentrations of larger particle sizes as compared to all other SLP-related AGPs.

**Loud phonation.** Loud phonation as an AGP was reported in one research study of moderate level of evidence. Investigation of the number and size distribution of particles produced in loud voicing revealed increased particle emission with loud phonation, however the distribution of particle size was independent of vocal loudness [27]. From this study, it can be concluded that a greater number of aerosolised particles are generated in loud voicing as opposed to voice production at lower volumes, however the act of voicing loudly does not seem to have an impact on the size of particles generated.

**Singing.** Singing was not explored in any of the research studies, with the highest level of evidence found across the four review articles and clinical guidelines exploring this AGP being in the very low category.

## Certainty of evidence

In the research studies category, the certainty of evidence as evaluated by GRADE ranged from very low to moderate, with ten of the 24 papers falling in the 'very low' category, six papers classed as 'low' certainty and eight as 'moderate'. Review studies ranged from very low to low certainty of evidence, with seven papers categorised as 'very low' and five as 'low' certainty of evidence. All three clinical guidelines fell within the category of 'very low' certainty of evidence.

AGPs identified across the research studies were coughing, breathing, speaking, singing and sustained phonation. The included review articles identified breathing, coughing, speaking, and singing as AGPs. The focus of the three included clinical guidelines was management of swallowing disorders and recent dysphonia, service delivery, clinical procedures, infection control, prevention of the transmission of COVID-19 and the use of PPE. AGPs mentioned in these guidelines included coughing, talking, singing, voice assessment tasks, forceful blowing and the Lee Silverman Voice Treatment [61].

## Evidence for risks of infectious disease transmission associated with identified AGPs

Risk of transmission of an infectious disease associated with specific aerosol-generating procedures was examined in both human-based and modelling studies. Table 4 presents the studies that examined the risk and mechanism of infectious disease transmission related to each of the identified AGPs. Disease transmission by cough was examined by the largest number of

**Table 4. Records examining risk of infectious transmission of aerosols across AGPs.**

| AGPs with Aerosol Infection Transmission explored | Number of articles exploring AGP infection transmission | Area of focus | Specific papers exploring AGP |
|---|---|---|---|
| Coughing | 12 | Infection risk model | [26] |
| | | Exposure to aerosol fluid | [43] |
| | | Viable virus within aerosol particles | [36, 37, 51] |
| | | Risk of COVID-19 or other respiratory disease transmission | [3, 15, 51–53, 57] |
| | | Impact of particle size on disease spread | [49] |
| Speaking | 8 | Viable virus within aerosol particles | [37, 51] |
| | | Risk of COVID-19 or other respiratory disease transmission | [3, 15, 51, 53, 57] |
| | | Impact of particle size on disease spread | [49] |
| Breathing | 6 | Viable virus within aerosol particles | [36, 37] |
| | | Risk of COVID-19 or other respiratory disease transmission | [52, 53, 57] |
| | | Impact of particle size on disease spread | [49] |
| Singing | 2 | Risk of COVID-19 or other respiratory disease transmission | [3, 55] |

studies, followed by speaking and breathing. For most of these AGPs, risk of transmission was assessed based on the following variables:

**Presence of a specific viable pathogen in aerosols.** Across the included studies, different viruses were investigated. Two research studies were specific to COVID-19, five were specific to influenza, and one was specific to MERS. Two studies used the term 'respiratory viruses' and 14 did not refer to a virus. Within the review category, eight papers were specific to COVID-19, one was specific to influenza, and three studies did not provide a specific pathogen. All three clinical guidelines were specific to COVID-19.

**Transmission capability of aerosol particles.** The transmission capability of aerosol particles generated by identified AGPs was investigated according to the following variables:

1. Particle size and distribution: Particles within different ranges of diameters behaved differently, which determined their airborne duration.

2. Type of AGPs: Infectious capability of aerosols and risk exposure according to each of the AGPs examined.

3. Environmental factors: This determined the dynamics of airflow, aerosol distance travelled and mode of transmission. Table 3 highlights studies that examined environmental factors related to aerosol transport characteristics across the AGPs of interest.

Four research studies identified in the review investigated or reported risk of transmission of viruses in aerosols. These studies referred to three activities: coughing, speaking and breathing. The highest certainty of evidence was found in coughing (moderate certainty of evidence), with the remaining activities found to have low or very low evidence certainty.

**Transmission risk from coughing.** Four research studies explored transmission risk from coughing. The certainty of evidence for these papers ranged from moderate (1 study) to low (2 studies) to very low (1 study). The characteristics explored by these papers pertaining to transmission risk were models of infection risk, exposure to aerosol fluid and the presence of viable virus within aerosol particles. The study of moderate evidence [31] found that infection risk increases with increased frequency of close and prolonged contact with cough aerosols,

while the lower-evidence studies found that multiple exposure pathways can lead to transmission, and that an increased volume and number of particles emitted during coughing occurs when a subject has influenza [37, 43]. Collectively, these studies seem to conclude that close, prolonged and/or frequent contact increases transmission risk of influenza, and that there are a range of pathways through which this virus may be transmitted.

**Transmission risk from speaking.**   One research study explored speaking-related transmission risk. This study was found to be of low evidence and explored the characteristic of viable virus from aerosol particles, reporting that virus-positive samples of influenza could be collected from individuals when talking [37]. While this low-evidence study finding should be interpreted with caution, it suggests that viral transmission of influenza can occur from speaking but it is untested as to whether the finding applies to COVID-19.

**Transmission risk from breathing.**   Two research studies explored breathing-related risk of transmission, with one being of moderate certainty of evidence [31] and the other of low evidence certainty. Both studies explored the ability of breathing particles to contain viable virus for aerosol transmission. The study of moderate evidence found that 42% of participants could produce aerosols containing viable influenza virus from exhalation. The low evidence study echoed this finding; reporting that viable virus traces could be detected in breathing samples. It may therefore be concluded that viral transmission of influenza strains can occur through breathing-related activities.

## Discussion

### Principal findings

Across the 39 eligible records included in this systematic review, six behaviours that occur in SLP assessment and/or intervention were identified as aerosol generating: coughing, speaking, breathing, singing, sustained phonation and loud voicing. Three types of literature were identified (research studies, review articles and clinical guidelines), with the findings of research studies forming the basis of the other two record types. Data extracted from research studies was typically of low-level evidence. Review articles used and applied this data to a wider range of AGPs, often without direct evidence for the same. Clinical guidelines then applied the research study evidence and review article findings with even broader application to risk assessment and harm minimisation contexts, so that a type of 'nesting effect' was observed (see Fig 2).

The collective evidence across all behaviours was highly varied in terms of parameters explored, outcomes measured and methodological approach. There is a lack of high-level evidence common to all AGPs in SLP clinical practice, with only at best, moderate levels of evidence supporting the aerosol-generating properties of each behaviour. There is evidently more research exploring coughing and its aerosol-generating potential than the other AGPs of interest. While the collective data seems to suggest coughing as the AGP of highest transmission risk, this behaviour has been investigated with greater frequency than others. It is unknown whether other AGPs occurring in SLP and community practice (e.g. singing) may produce an equivalent, lesser or higher transmission risk, as these activities exist in domains where far less empirical research is conducted.

### Strengths and limitations of the study

Records included in the systematic review covered all three types of literature available at the time of the study, including original research studies, review articles, and clinical guidelines. The data extraction process followed specific guidelines (STROBE for research studies, PRISMA for reviews, and RIGHT for clinical guidelines) so that essential and comprehensive

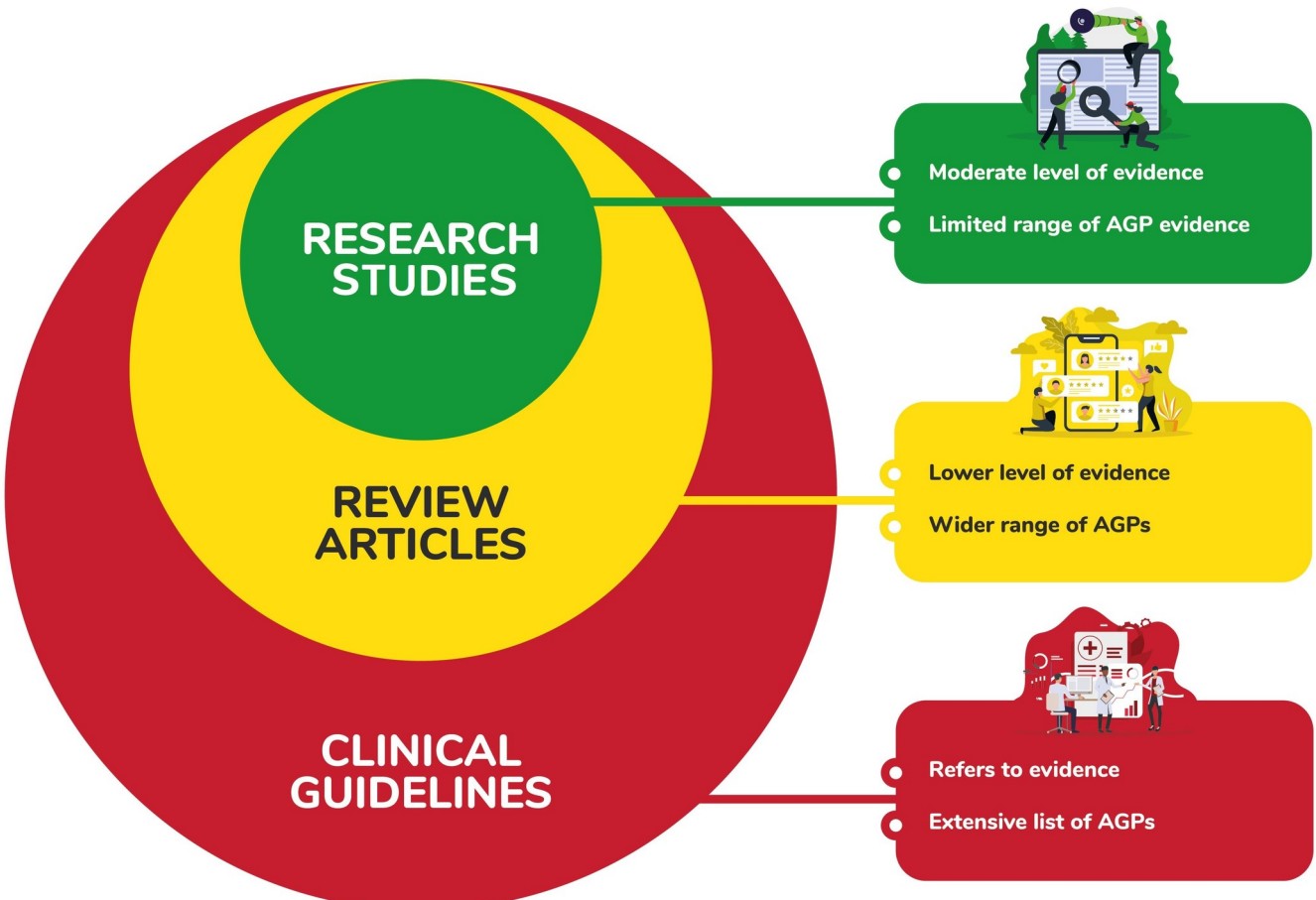

**Fig 2. Visual representation of nesting effect across article types.**

information was obtained. Limitations of this study included a lack of comprehensive review of the means to mitigate risk (time/ resource restrictions) and therefore no critical evaluation of these. This study only included studies published in English. Due to a lack of data and heterogeneity across measurement methods, we could not calculate relative risks of infectious transmission associated with the different AGPs in focus. Finally, the paucity of robust data available regarding COVID-19 and SLP clinical tasks limits our ability to generalise these findings to the current pandemic. As such, we are unable to develop well-informed recommendations and conclusions pertaining to SLP COVID-19 management and abatement of risk, as these conclusions would not be supported by research we would describe as either reliable or accurate.

## Comparison with other studies

The results of this systematic review are similar to those found in reviews of AGPs in dysphagia. The current review found a lack of consensus, high risk of selection bias and focused only on risk of infection based on previously-identified AGPs in the literature [15]. A similar trend was identified in a recent review exploring dysphagia and COVID-19; finding that no high-quality scientific papers had been published to guide the recommendations being advised of clinicians globally [16]. This dysphagia paper reviewed only three articles which specifically

explored dysphagia management in patients infected with COVID-19, compared to the 39 records reviewed in the present study.

The deficiency in well-reported, direct and high-certainty evidence research in the context of the pandemic has likewise been echoed by research in respiratory physiology and ventilation [62]. Similarly, a recent rapid review of AGPs in the context of clinical guidance for dental practitioners [63] noted limited evidence to support the majority of recommendations in the reviewed guidance documents. Across these studies overall, it is evident that the quality of research and research design in examining aerosol generation and COVID-19 in health is often lacking. However, in the current climate of the pandemic, the need for HCW protection seems to have necessitated the use of evidence out of context and possibly in some cases, inappropriately.

## Re-naming specific AGPs associated with SLP as aerosol-generating behaviours

We propose a change to the terminology surrounding AGPs to better describe those non-invasive clinical and day-to-day tasks that are reported to result in aerosol generation. As such, we suggest that the term 'aerosol-generating procedures' be limited to describing those medical/surgical processes occurring in a hospital, or medical-based setting, while the term 'aerosol-generating behaviours' (AGBs) be applied to those behaviours that generate aerosols in broader settings, including clinical SLP assessment and intervention. These 'behaviours' can further be categorised into vegetative acts (such as coughing and breathing), verbal communication activities of daily living (such as speaking and loud voicing) and performance-based tasks (such as singing and sustained phonation).

## Implications for research and future studies

Unfortunately, the combined results of this systematic review do not allow us to provide guidance to speech language pathologists about the relative risk of various AGBs. This is partly due to the combination of limited scope and diversity of topic of each of the papers included. Additionally, much of the research included was obtained in laboratory settings which, while important, inadequately represents clinical practice. Clearly, well-designed and controlled research is required to address this lack of robust data across all AGBs explored, ideally with the use of clinical comparators and further mathematical modelling to establish a SLP-specific risk matrix. Further research is also needed to provide higher levels of evidence for management strategies in reducing the risk of aerosol infectious disease transmission, particularly in the age of COVID-19. In summary, two significant recommendations emerge; 1) a clear need for relative risk ratios between the behaviours to be examined, and 2) SLPs currently need to treat all these behaviours as high risk.

This systematic review, whilst focused on SLP practice, has investigated the evidence for activities that occur in broader healthcare settings and common activities of daily human functioning, specifically verbal communication. Results can therefore be extrapolated appropriately to physical face-to-face interactions in other healthcare contexts and equivalent verbal interactions, which occur frequently throughout the wider global community.

## Supporting information

**S1 Checklist. PRISMA 2009 checklist.**
(PDF)

**S1 File. Infographic V3.**
(PDF)

**S2 File. Search strategy.**
(PDF)

**S3 File. Data extraction tables.**
(PDF)

**S4 File. PROSPERO updated record V2.**
(PDF)

## Author Contributions

**Conceptualization:** Patricia McCabe, Catherine Madill.

**Data curation:** Antonia Margarita Chacon, Duy Duong Nguyen.

**Formal analysis:** Antonia Margarita Chacon, Duy Duong Nguyen.

**Investigation:** Antonia Margarita Chacon.

**Methodology:** Antonia Margarita Chacon, Patricia McCabe.

**Project administration:** Antonia Margarita Chacon.

**Resources:** Catherine Madill.

**Validation:** Antonia Margarita Chacon.

**Visualization:** Antonia Margarita Chacon.

**Writing – original draft:** Antonia Margarita Chacon.

**Writing – review & editing:** Antonia Margarita Chacon, Duy Duong Nguyen, Patricia McCabe, Catherine Madill.

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
