## [Decision Letter · Decision Letter 0]

25 Mar 2021

PONE-D-20-37986

Aerosol-generating behaviours in speech pathology clinical practice: a systematic literature review

PLOS ONE

Dear Dr. Chacon,

Thank you for submitting your manuscript to PLOS ONE. After careful consideration, we feel that it has merit but does not fully meet PLOS ONE’s publication criteria as it currently stands. Therefore, we invite you to submit a revised version of the manuscript that addresses the points raised during the review process.

Please address Reviewer 2's critique and include data published outside of the speech-language pathology literature including research generated by physicists, engineers, and other healthcare professionals.

We look forward to receiving your revised manuscript.

Kind regards,

Marie Jetté

Academic Editor

PLOS ONE

Journal Requirements:

1. Please ensure that your manuscript meets PLOS ONE's style requirements, including those for file naming. The PLOS ONE style templates can be found athttps://journals.plos.org/plosone/s/file?id=wjVg/PLOSOne_formatting_sample_main_body.pdf andhttps://journals.plos.org/plosone/s/file?id=ba62/PLOSOne_formatting_sample_title_authors_affiliations.pdf

Additional Editor Comments (if provided):

Please address Reviewer 2's critiques and include data generated by physicists, engineers, and other healthcare practitioners.

Reviewers' comments:

Reviewer's Responses to Questions

**Comments to the Author**

1. Is the manuscript technically sound, and do the data support the conclusions?

Reviewer #1: Yes

Reviewer #2: Partly

2. Has the statistical analysis been performed appropriately and rigorously? 

Reviewer #1: N/A

Reviewer #2: Yes

3. Have the authors made all data underlying the findings in their manuscript fully available?

Reviewer #1: Yes

Reviewer #2: Yes

4. Is the manuscript presented in an intelligible fashion and written in standard English?

Reviewer #1: Yes

Reviewer #2: Yes

5. Review Comments to the Author

Reviewer #1: This review is timely and highlights the unique expertise of speech-language pathologists in respiratory-related activities during a respiratory virus pandemic. I believe the authors have done a notable job in describing their methods carefully and also using caution with interpreting their findings. I particularly appreciate the proposed distinction between aerosol-generating procedures vs. behaviors (and further by vegetative, ADLs, and performance). In addition, an advantage of this review is that the authors also highlighted how other professionals and scientists will find this information useful going forward during the era of COVID-19.

Reviewer #2: Although this manuscript follows a strong methodological protocol and is well written, it relies on two fundamental misconception, which limit is scientific relevance:

1. The CDC defines AGPs as those medical procedures that are “more likely to generate higher concentrations of infectious respiratory aerosols than coughing, sneezing, talking, or breathing." Naming these physiological phenomena as aerosol generating behaviors within the context of SLP practice is not sound, as it implies that there is something unique about SLP practice making these phenomena riskier than in other healthcare settings or in the community.

2. Focusing on the body of evidence on aerosol generation in relation to speech pathology ignores the high quality data produced by physicists, engineers, other healthcare practitioners on aerosol generation during speech, singing and coughing. It is not clear to me why the authors decided to exclude this data and only focus on the speech pathology literature, as it ignores the multidisciplinary efforts towards better understanding of aerosol generation going much beyond the speech literature.

6. PLOS authors have the option to publish the peer review history of their article (what does this mean?). If published, this will include your full peer review and any attached files.

Reviewer #1: No

Reviewer #2: No

---

## [Author Response · Author response to Decision Letter 0]

30 Mar 2021

REVIEWER 1 RESPONSE: 

 Thank you for your comment. 

REVIEWER 2 RESPONSE: 

Thank you for your comment and careful evaluation of our paper. 

1. Thank you also for providing the CDC definition of AGPs. While the CDC’s definition focuses upon those procedures eliciting higher concentrations of infectious respiratory aerosols than coughing, sneezing, talking or breathing, given that these activities do generate infectious respiratory aerosols and are routinely performed in our clinical practice, we wished to examine these specific behaviours more closely from a Speech Pathology perspective. The CDC definition appears to exclude these behaviours being considered as AGPs despite these activities occurring routinely in SLP practice and daily life. Therefore, in introducing the concept of AGBs, we wished to further elucidate the risk posed by performing these behaviours in the context of healthcare practice and societal interactions. 

You have mentioned that we imply that there is something unique about SLP practice which makes these phenomena riskier than in other contexts. We agree that there are components of our practice which are in fact unique to other contexts in which the behaviours of breathing, speaking, singing, coughing, loud voicing and sustained phonation are produced. In SLP assessment and therapy sessions, the clinician and patient typically sit in an enclosed space for an hour, purposefully producing these tasks which are likely to generate high levels of aerosols in close proximity. There are no other contexts of which we are aware where this range of behaviours with aerosol-generating potential are elicited purposefully and routinely across interactions. While we cannot state conclusively that the tasks performed in SLP clinical contexts make the practice of speech pathology riskier, we think it is reasonable to assume there is some degree of higher risk owing to the length of time of SLP assessment and treatment sessions, and specific, often repeated cueing of these behaviours that occurs in these sessions. 

As reviewer 1 has acknowledged in their comment ‘an advantage of this review is that the authors also highlighted how other professionals and scientists will find this information useful going forward during the era of COVID-19’, we have not limited the application of our findings to a SLP context but rather, aimed to extrapolate our conclusions to broader settings, healthcare and otherwise. 

2. We absolutely agree that solely exploring a speech pathology context would not enable us to draw well- informed conclusions regarding the aerosol generating potential across the six explored behaviours.

We apologise if it was unclear from our paper that we did in fact explore these behaviours across a range of fields beyond speech pathology. If we bring the reviewer’s kind attention to our search strategy (supplementary material S2), page 1 (‘Speech Pathology Clinical Tasks’)- the terms used to explore aerosol generation in SLP would not have excluded non-SLP literature. The terms searched include such terms as ‘breath* OR cough* OR speech*…’, which were then coupled with other terms (‘coronavirus’, ‘aerosols’ and ‘transmission risk’) across our database searches, which certainly did yield papers from the fields of engineering, other healthcare settings, etc. as has been recommended. 

We also acknowledge an additional area requiring clarification is that our exclusion criteria specified in pages 8-9 of the manuscript detailed that one of our exclusion criteria was papers ‘not relevant to speech pathology voice, resonance or motor speech clinical practice’. We understand how this may have been misconstrued to suggest we excluded papers that weren’t written specifically for speech pathology, but in fact meant to indicate that any articles not exploring the tasks of interest (speech, breathing, singing, sustained phonation, loud voicing and prolonged phonation) were excluded, i.e. no exclusion occurred if articles were not specifically related to the SLP field. 

We have therefore amended the inclusion criteria (pages 8-9) to clarify that papers not exploring the behaviours of interest were excluded; not exclusion of papers beyond the field of SLP. 

As further evidence of this, we’d like to kindly refer the reviewer to a further supplementary materials document uploaded with our submission (S3), to examine the titles, outcome measures and findings of included studies. In reviewing this list, we hope it becomes clear that there is a limited number that were specific to speech pathology (only Giovanni et al., 2020 and two ‘clinical guidelines’ documents), with the remainder being conducted across broader multidisciplinary settings. 

Finally, upon initial submission of our paper, we were requested by PLOS One to modify the title of our review from ‘Aerosol-generating behaviours in voice and speech: A systematic literature review’ to a title that was more ‘specific, descriptive, concise, and comprehensible to readers outside the field (for example by specifying that the systematic review focuses on AGPs in speech pathology clinical practice)’. As such, we framed our paper title to focus moreso upon speech pathology following this request by the editorial board, despite our intention that our paper explore and be applied to contexts beyond the field of SLP.

---

## [Editor Report · Decision Letter 1]

5 Apr 2021

Aerosol-generating behaviours in speech pathology clinical practice: a systematic literature review

PONE-D-20-37986R1

Dear Dr. Chacon,

We’re pleased to inform you that your manuscript has been judged scientifically suitable for publication and will be formally accepted for publication once it meets all outstanding technical requirements.

Kind regards,

Marie Jetté

Academic Editor

PLOS ONE

Additional Editor Comments (optional):

Thank you for responding to the reviewers' suggestions.
---

## [Editor Report · Acceptance letter]

16 Apr 2021

PONE-D-20-37986R1 

Aerosol-generating behaviours in speech pathology clinical practice: a systematic literature review 

Dear Dr. Chacon:

I'm pleased to inform you that your manuscript has been deemed suitable for publication in PLOS ONE. Congratulations! Your manuscript is now with our production department. 

Kind regards, 

on behalf of

Dr. Marie Jetté 

Academic Editor

PLOS ONE